# Investigation of the Turbulent Drag Reduction Mechanism of a Kind of Microstructure on Riblet Surface

**DOI:** 10.3390/mi12010059

**Published:** 2021-01-06

**Authors:** Mingrui Ao, Miaocao Wang, Fulong Zhu

**Affiliations:** 1School of Materials Science and Engineering, Huazhong University of Science and Technology, 1037 Luoyu Road, Wuhan 430074, China; aomingrui@hust.edu.cn; 2Institute of Microsystems, School of Mechanical Science and Engineering, Huazhong University of Science and Technology, 1037 Luoyu Road, Wuhan 430074, China; blakewangmc@163.com

**Keywords:** turbulence intensity, vortex, microstructure, drag reduction rate, frictional resistance

## Abstract

With the k-ε renormalization group turbulence model, the drag reduction mechanism of three- dimensional spherical crown microstructure of different protruding heights distributing on the groove surface was studied in this paper. These spherical crown microstructures were divided into two categories according to the positive and negative of protruding height. The positive spherical crown micro-structures can destroy a large number of vortexes on the groove surface, which increases relative friction between water flow and the groove surface. With decreasing the vertical height of the spherical crown microstructure, the number of rupture vortexes gradually decreases. Due to the still water area causes by the blocking effect of the spherical crown microstructure, it was found that the shear stress on the groove surface can be reduced, which can form the entire drag reduction state. In another case, the spherical crown microstructures protrude in the negative direction, vortexes can be generated inside the spherical crown, it was found that these vortexes can effectively reduce the resistance in terms of pressure and friction. In a small volume, it was shown that the surface drag reduction rate of spherical crown microstructures protrudes in negative directions can be the same as high as 24.8%.

## 1. Introduction

Nowadays microfluidics has been used in a wide range of areas, such as health detection [1], chemical pharmacy [2,3], precision sensors [4], etc. It is important to investigate the property of microfluidics.

It is often found that using microstructure can provide better performance improvement [5]. Past researches showed the resistance of a moving object at high speed in a fluid is largely affected by the flow resistance of the object. The frictional resistance between the surface of Airbus A340–300 and the air takes up to 48% in total resistance [6], which reduces the energy efficiency. In order to reduce energy loss from the friction of a fluid, some microstructures have been proposed to reduce the flowing resistance. In the 1980s, it was first discovered that the non-smooth groove structures on the skin of sharks have a positive effect on the surface drag reduction [7]. Since then, scientists from different disciplines have been investigating the microstructure surfaces inspired by some fast-swimming animals [8], and studying the drag reduction mechanism of that microstructure.

In recent years, micro-riblets on the skin of Shark were found to be effective in the reduction of flowing resistance. It was observed that these riblets on the surface of objects can hinder the horizontal translation of vortexes through the surface, and reduced vortex jets and outer turbulence [9,10] in the turbulent boundary layer of the surface. For these riblets on the surface, the outer side decreases the exchange of momentum from the twisting of vortexes in the outer layer of the boundary layer, and reduces the turbulent energy of lateral flow vortexes [11]. Similarly, the riblet microstructure on the surface of objects can lift vortexes off the surface and keep them above the riblet tip. As a result, the area of the surface with riblet microstructure which exposes in high-speed flowing fluid is limited, and the turbulent flowing energy at the bottom of the riblet plate is significantly lower than that of the whole riblet plate on the surface. Subsequently, a laminar layer is formed in the riblet plate on the surface. It was showed that many factors are beneficial to reducing the surface shear stress between flowing fluid and the surface structure of objects [12,13,14].

After decades of continuous exploration, the surface drag reduction technology using bionic surface microstructure has been applied in many fields. Choi et al. [14] studied the hydrodynamic mechanism of drag reduction with V-shaped grooves with numerical simulation methods, and Figure 1 shows three kinds of micro-groove models. Lee [13] focused on the wall flow state of a turbulent boundary layer in a semicircular groove, and Samni et al. [15] also simulated the blade-shaped groove surface against a low Reynolds number turbulent groove. From these results, it was found that the drag reduction rate of the trench could reach a maximum of 11% when the trench height and clearance ratio is around 0.5. In the research work on the fan-shaped grooves with different h/s, Bechert et al. [16] observed that the best rate of drag reduction approximates to 6.5% when the scale of h/s is 0.7.

Due to their simple lateral interface structure, some typical two-dimensional groove surfaces, such as V-shaped grooves, fan-shaped grooves, and blade-shaped grooves are selected to thoroughly study surface microstructure models of their grooves in the flowing direction [10]. However, the drag reduction rate of these surface microstructure models is not very prominent. In order to improve the surface drag reduction rate, the effect of surface hydrophobicity is proposed. Neihuis et al. [17,18] studied the surface microstructures of 300 kinds of plants, found spherical crown structures and waxy substances of micro-scale exist on the rough surface of plants. The surface microstructure of plant surfaces shows obvious hydrophobicity. Watson et al. [19] studied the microstructure of termite and cicada wings, and found spherical crown-like protrusions are distributed on their wings. Those microstructures of spherical crown-like protrusions can be proposed in a new direction to improve the surface drag reduction rate.

In this paper, a new model combining the traditional V-shaped groove surface with a uniformly distributed spherical crown structure was proposed, the turbulent drag reduction mechanism of these microstructures was analyzed in the following paper.

## 2. Materials and Methods

### 2.1. Theoretical Calculation Methods

It is generally believed that the valley bottom of the groove can effectively limit the turbulence of the fluid flows and reduce the energy exchange between the fluid and the wall, the V-shaped groove can achieve a good drag reduction effect. However, due to the vortices that formed from the fluid flows through the tip of the V-shaped groove, the drag reduction effect of the entire structure will weaken. In this work, the RNG k-ε model was used to accurately simulate the influence of the vortices at the bottom of the groove.

The RNG k-ε is a model derived from the instantaneous Naiver-stokes equation with the method of the re-normalization group [20]. Compared with other turbulence models, the RNG k-ε model has the following advantages:The RNG k-ε model has a more accurate description of the dissipation rate of turbulent kinetic energyThe RNG k-ε model considers the influence of eddy currents on turbulence and improves the accuracy of vortices flowThe RNG k-ε model provides an analytical formula for the turbulent Prandtl number, which can more accurately simulate the problem of turbulent boundary layer flow.

The N-S equation is expressed as [21]:(1)$$\frac{\partial }{\partial t}\left(\rho u_{i}\right)+\frac{\partial }{\partial x_{j}}\left(\rho u_{i}u_{j}\right)=-\frac{\partial p}{\partial x_{i}}+\frac{\partial \tau_{ij}}{\partial x_{j}}+\rho f_{i,}$$
where *ρ* is the fluid density, *p* is hydrostatic pressure, *u_i_* and *u_j_* are velocity component of a fluid element, *τ_ij_* is sheer stress, *f_i_* is unit mass force, *x_i_* and *x_j_* are components of displacement, *t* is the time.

The RNG *k-ε* model has an improvement based on the standard *k-ε* equation. This equation has more components to be analyzed and the effect of turbulent vorticity is considered.

In the RNG *k-ε* model, the turbulent kinetic energy equation (*k* equation) is [21]:(2)$$\frac{\partial }{\partial t}\left(\rho k\right)+\frac{\partial }{\partial x_{i}}\left(\rho ku_{i}\right)=\frac{\partial }{\partial x_{j}}\left[\left(\mu_{0}+\frac{\mu_{t}}{\sigma_{k}}\right)\frac{\partial k}{\partial x_{j}}\right]+P_{k}+\rho_{\varepsilon ,}$$
where *k* is turbulent kinetic energy, *μ_t_* is molecular viscosity coefficient, *P_k_* is turbulence generation term.

The turbulent energy dissipation rate equation (ε equation) is [21]:(3)$$\frac{\partial }{\partial t}\left(\rho \varepsilon \right)+\frac{\partial }{\partial x_{i}}\left(\rho \varepsilon u_{i}\right)=\frac{\partial }{\partial x_{j}}\left[\left(\mu_{0}+\frac{\mu_{t}}{\sigma_{\varepsilon }}\right)\frac{\partial \varepsilon }{\partial x_{j}}\right]+C_{\varepsilon 1}\frac{\varepsilon }{k}P_{k}-C_{\varepsilon 2}\rho \frac{\varepsilon^{2}}{k}-R_{\varepsilon ,}$$
where the turbulent viscosity coefficient is
(4)$$\mu_{t}=\rho C_{\mu }\frac{k^{2}}{\varepsilon },$$
where RNG *k-ε* model-specific tuning parameters:(5)$$R_{\varepsilon }=\frac{C_{\mu }\rho \eta^{3}(1-\eta /\eta_{0})}{1+\beta \eta^{3}}\frac{\varepsilon^{2}}{k}.$$

These constants in Equations (2) and (3): *σ**_k_* = 1.39, *σ_ε_* = 1.39, *C_ε_*_1_ = 1.42, *C_ε_*_2_ = 1.68, *C_μ_* = 0.0845, *η*_0_ = 4.38, *β* = 0.012 [22].

### 2.2. Surface Microstructure Modeling

In this paper, a special groove microstructure was selected for investigating the surface drag reduction mechanism, which is a number of spherical crowns set on the surface of the V-shaped groove. This special groove microstructure with the crown was improved from a typical V-shaped groove similar to the triangular groove structure of shark skin size.

The dimensions of the V-shaped groove used in this paper were measured by other researchers [23]. Based on these measurements, an averaging process was conducted. In the following of this paper, the depth “h” of a special V-shaped groove is 16μm, and the width “s” of a special V-shaped groove is 48 μm. To avoid overly damaging basic V-shaped groove from the spherical crown, the radius of the spherical crown on the surface of the special V-shaped groove is set to be 4 μm in this paper. Bechert et al. [9] discovered staggered three-dimensional blade grooves could be a better function of drag reduction. In this paper, the layout of spherical crowns on the surface of V-shaped grooves is considered in modeling the special groove structure. The model of spherical crowns with a radius of 4 μm are staggered and uniformly distributed on the surface of special V-shaped grooves at a pitch of 30 μm. The height of the complete crown is defined as r, and the r is 4 μm. A series of special V-shaped grooves are formed by changing the height of spherical crowns, and Figure 2 shows ten different heights of crowns on the surface of special V-shaped grooves. Each height of the crown constitutes a three-dimensional spherical crown microstructure. When modeling, those bulge-like spherical crowns with positive protrusion height are set to be positive, and the concavity like spherical crowns with negative protrusion height is set to be negative at the same time.

The flow region illustration in modeling about the special groove microstructure with spherical crowns shows in Figure 3, coordinate axes Z, Y are the downstream direction and vertical entry direction respectively, X is the direction of expansion of V-shaped groove surface, and the scale of X:Y:Z is 288:640:900.

The flow region with two different surface shows in Figure 3. The entire flow region can be approximately regarded as a hexahedron, all boundary conditions can be set in the six directions of the hexahedron [24]:

(1) The upper wall and the lower wall of this flow region are set as smooth surface, because these two surfaces simulate the groove surface which needs to be smooth.

(2) Two surfaces in the Z-direction are set as the inlet surface and the outlet surface. The boundary condition at the inlet surface is set as a velocity condition, and the outlet surface is set as free flow.

(3) The last two surfaces which perpendicular to the flow direction are set as the symmetrical boundary.

This model simulates a small region in the entire flow field, using a symmetrical boundary can ensure that the simulated region reflects the characteristics of the entire flow field.

In order to evaluate the effect of V-shaped groove surface with a spherical crown on the drag reduction mechanism, different flowing velocity is selected to calculate and analyze with a finite element method, while the height of spherical crowns on V-shaped groove surface would be changed. So, the turbulent kinetic energy and surface shear stress on this grooved surface can be obtained and explored in detail [25].

## 3. Results and Discussion

In this paper, it is focused on discriminate the influence of the spherical crown groove structure with different heights on the overall drag reduction rate of the surface. The turbulent kinetic energy and the surface shear stress on the V-shaped groove surface with a spherical crown of different height can be achieved, the drag reduction mechanism of positive spherical crowns and negative spherical crowns will be respectively discussed, and some difference about the drag reduction mechanism of two models can be defined.

### 3.1. Positive Spherical Crown Structure

The drag reduction rate of the traditional V-shaped groove model from the shark skin surface is generally about 6.5%, which is undesirability. Based on this, the two-dimensional V-shaped groove structure is partially modified in this model, which distributes positive spherical crowns of different heights on the surface of the V-shaped groove. For the spherical crown structure of the V-shaped groove surface, when the protruding direction of the spherical crown is positive, the curve of the drag reduction rate with the flow speed is shown in Figure 4. From this curve, the characteristic of drag reduction is a very obvious correlation with the water flow velocity on the surface of a positive spherical crown structure. For a V-shaped groove structure with a positive crown microstructure, it has some small drag reduction effects in low flow speed. However, from the curve of Figure 4, it can be observed that a V-shaped groove structure with positive crown microstructure behaves counteraction for flowing liquid with a high speed, in this case, those positive crowns produce and increase the resistance of flowing liquid.

The drag reduction potential of the groove is determined by the effective penetration height of the crowns on the surface [10]. Figure 5 shows the situation with a complete crown. Figure 5a is a contour of velocity near the groove surface. In this contour, it can see that there are the same burst vortexes on the top of the crown. Figure 5b iconifies the turbulent kinetic energy on the groove surface, it can be seen that the position of large peaks coincides with the position of triangular groove, and the two small tips on each large peak coincide with the position of two protruding spherical crowns on the surface of V-shaped groove structure. From the contour of velocity and curve of turbulent kinetic energy near the groove surface, it can be determined that those positive spherical crowns cause a burst of flow vortexes, which result in a large number of secondary vortexes. These vortexes increase the interaction between the fluid and the surface of the wall, which induces an increase in the frictional resistance of the wall surface.

In addition, due to the blocking effect of the spherical crown on the water flow, there is a still water area within a certain range behind the crown, which shows in Figure 6 and Figure 7, the flow velocity in this still water area decreases sharply, and the surface shear stress also decreases accordingly. Because of the low turbulent kinetic energy, it can be judged that the turbulent motion of water stabilizes than another place. In this area, the frictional resistance plays a key role, and the wall shear stress here is obviously reduced, which decreases frictional resistance.

In Figure 6 and Figure 7, the drag reduction rate of the positive spherical crown is determined by both the pressure resistance from the blocking of flow and the frictional resistance which performance as the area of the still water area. When the flow velocity becomes slow, the frictional resistance begins to be the main factor affecting the drag reduction rate. In this situation the still water area is always large, so a V-shaped groove structure with a positive spherical crown has relatively higher drag reduction. When the flow velocity increases, the effective height of the spherical crown plays a crucial role in blocking the vortex on the surface, which increases the pressure resistance, and the drag reduction rate of the groove structure enters into the stage of increasing flowing resistance in Figure 4.

The influences of different height of positive spherical crown on V-shaped groove surface have some difference for the flow velocity and the drag reduction rate, Figure 8 shows the relationship between groove surface and the drag reduction ratio of spherical crowns with different positive protruding heights. The different height of positive spherical crowns mainly affects the ability of the groove to breaking vortexes, and its specific manifestation is the intensity of the turbulent kinetic energy on the groove surface. Stronger the turbulent kinetic energy, the greater the friction between broken vortexes and groove wall surface, and the greater its friction resistance is. The prominent turbulent kinetic energy generates at the top position of the complete hemispherical crown structure, which shows in Figure 9a, indicates that there are a large number of broken vortexes. these vortexes can increase the frictional resistance on the surface. As crowns’ height decreases, the turbulent kinetic energy shows in Figure 9 decreases, which means there are not many broken vortexes to create frictional resistance. The still water area behind the crown dominates the drag reduction processing of the V-shaped groove structure with a positive spherical crown, so the drag reduction rate of the groove can be gradually increasing.

### 3.2. Negative Spherical Crown Structure

To compare with the positive spherical crowns’ microstructure, the drag reduction effect of the V-shaped groove structure with the concavity microstructure named negative spherical crown is explored. When the protruding spherical crown on the surface of the V-shaped groove becomes concavity as the negative spherical crown, the drag reduction effect is completely different from the positive spherical crowns’ microstructure at the same flowing speed. The curve of the drag reduction rate with the flow velocity shows in Figure 10. It is very clear that the drag reduction rate decreases with the speed increasing, and has positive dependence on the flowing velocity. Comparing with the groove surface of the positive crown, the value of the drag reduction rate of the V-shaped groove structure with the negative spherical crown is much higher at the same flowing speed. In Figure 10, when the flow velocity approaches 8 m/s, the drag reduction rate is close to the maximum of 25%. From a general point of view, after this maximum value, the drag reduction rate decreases with the flow velocity increasing but still maintains a high drag reduction rate. The drag reduction rate of a V-shaped groove structure with a negative spherical crown far exceeds the conventional V-shaped groove.

In order to investigate the drag reduction effect of the negative spherical crown on the groove surface, an observation line which the position as same as Figure 11 is set along the direction of the water flow, and it is used to observe various parameters of the water flow after passing through the negative spherical crown. The way negative spherical crowns affect the flow vortexes near the surface is very different from the traditional V-shaped groove surface. The intensity of turbulent kinetic energy and water flow velocity is observed by the observation line. It can be clear that vortexes play an important part in the drag reduction effect. Figure 11 shows the vortex at the location of the negative crown microstructure on the V-shaped groove surface. When the water with some speed flows through the negative spherical crown, vortexes form in the spherical crown. A part of the water flows through vortexes and reflows with the main fluid. When two parts of the fluid combine to a flow velocity peak, which shows in Figure 12a, the turbulent kinetic energy can also be used to comprehend the state of flowing water on the surface of the negative crown microstructure. Figure 12b translates the relationship between turbulent kinetic energy with the flow direction. Combining Figure 11 and Figure 12, in the position of the negative spherical crown, the flow velocity roses sharply and the turbulent kinetic energy has a sharp decline. The sharply decreasing turbulent flow energy shows the water flow near the negative spherical crown has been converted from turbulent to laminar. This mechanism plays a crucial role in the drag reduction effect of negative crown microstructure.

In the field of hydrokinetics, it is well known that the wall surface resistance is mainly composed of two aspects of friction of vortexes with the surface and the pressure resistance of the water with the surface. Because of the existence of riblets, vortexes locate above the groove, interacting only with the peak of the grooved surface, and lateral flow velocity pulsation in the groove is much smaller than the lateral flow of riblets’ top [13]. In this paper, the flowing water in the groove valley is relatively close to the state of laminar flow. Due to the existence of a negative spherical crown, the momentum exchange at the bottom of the groove valley further reduces, which led to the reduction of friction resistance on the groove surface. In addition, vortexes form in negative spherical crown effectively convert the sliding friction between water flow and the wall surface into rolling friction, which also reduces the friction resistance from another aspect.

It is generally believed that the increase in wetting area in the groove surface structure increases the pressure resistance [10], but this mechanism has been changed due to the negative crown microstructure on the surface of the V-shaped groove. It is also mentioned earlier that the turbulent kinetic energy on the surface with a spherical crown indicates the boundary layer is conformable to the laminar structure. According to Bernoulli’s theorem, for the flowing water which is on the near-surface, the higher the velocity, the lower the pressure. In the situation of the negative crown, the pressure resistance is much smaller than the theoretical growth pressure resistance which is caused by the increase in the effective contact area. So, the negative spherical crown microstructure on the surface of the V-shaped groove effectively reduces the resistance from both the frictional resistance and the pressure resistance. Because of this, the drag reduction effect of the negative crown microstructure is beyond expectations in this paper.

The negative spherical crown microstructure of the V-shaped groove structure has a very clear effect of drag reduction, and Figure 13 shows the relationship between the flow velocity and the drag reduction rate on the groove surface of the negative spherical crown structure with different depths. The negative protrusion depth of the spherical crown mainly affects the size of the vortex created inside the crown. The interaction between vortexes and negative spherical crown is rolling friction, which is much smaller than sliding friction. As the depth of the crown gradually decreases, In Figure 14 it can be seen that the size of vortexes gets smaller following the decrease of the depth of the crown. When the depth of the crowns decreases to a critical depth just like Figure 14e, vortexes could completely disappear. During this process, the rolling friction of vortexes also transform to sliding friction. This transformation increases the friction between the water flow and the wall. As for the pressure resistance of the water with the surface, the pressure increases when the depth of the negative crown decreases. The contour of the pressure of the surface shown in Figure 15 indicates the increase of drag reduction rate of negative crown V-shaped groove microstructure.

## 4. Conclusions

In this paper, the flow characteristics of water in different V-shaped groove surface structures with the spherical crown microstructure was investigated. The effect of different spherical crown structures on drag reduction rates was analyzed by comparing the surface Reynolds shear stress, fluid velocity, near-wall turbulent kinetic energy and the distribution of intensity of vortexes. it leads to the following conclusions:

(1). Both the positive and negative protrusions of the spherical crown structure contribute similarly to reducing surface resistance, but there are also some differences. Two kinds of protrusions have a higher drag reduction effect for the flowing liquid with a low speed, and the drag reduction rate of two microstructures decreases with the increasing speed of flow water.

(2). The drag reduction effect of the negative spherical crown is generally better than the positive spherical crown microstructure.

(3). The microstructure of the spherical crown is protruding positive, the surface drag reduction rate of the spherical crown which the crown height +4μm is the lowest. As the height of the spherical crown decreases, the drag reduction rate of the groove surface also increases.

(4). The structure of the spherical crown is protruding negative, the surface drag reduction rate of the spherical crown which the crown height −4μm is the highest. As the height of the spherical crown increases, the drag reduction rate of the groove surface also increases. The negative spherical crown structure has good surface drag reduction characteristics, and the highest drag reduction rate can reach 24.8%.

## Figures and Tables

**Figure 1 micromachines-12-00059-f001:**
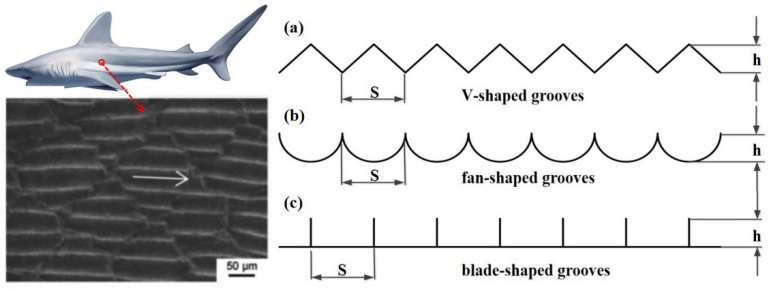
The microstructure of riblet on shark skin [5] and three kinds of micro-groove models.

**Figure 2 micromachines-12-00059-f002:**
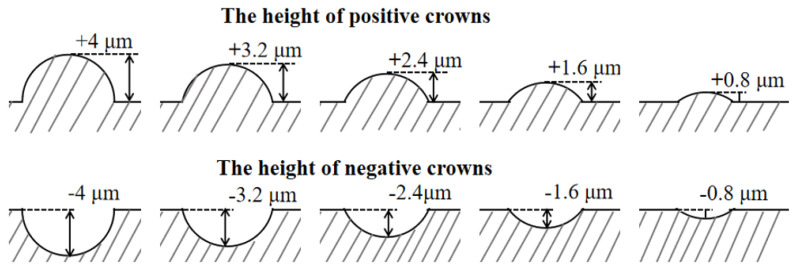
Ten different heights of spherical crowns used in this paper show in this illustration, the heights of these crowns are from +4 μm to −4 μm. Each height of the crown constitutes a three- dimensional spherical crown microstructure.

**Figure 3 micromachines-12-00059-f003:**
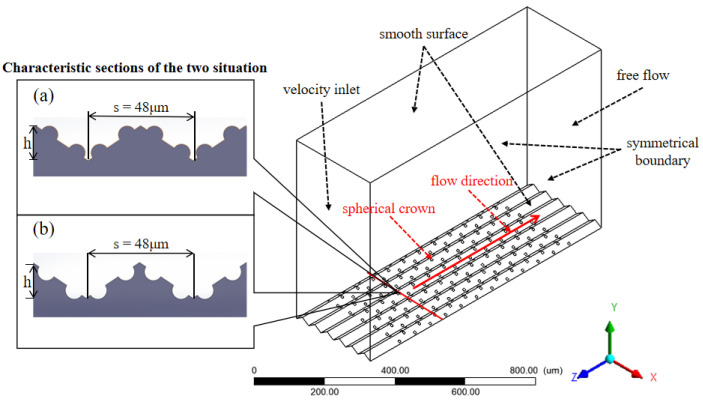
Flow region with two different surfaces; (**a**) positive crowns, (**b**) negative crowns.

**Figure 4 micromachines-12-00059-f004:**
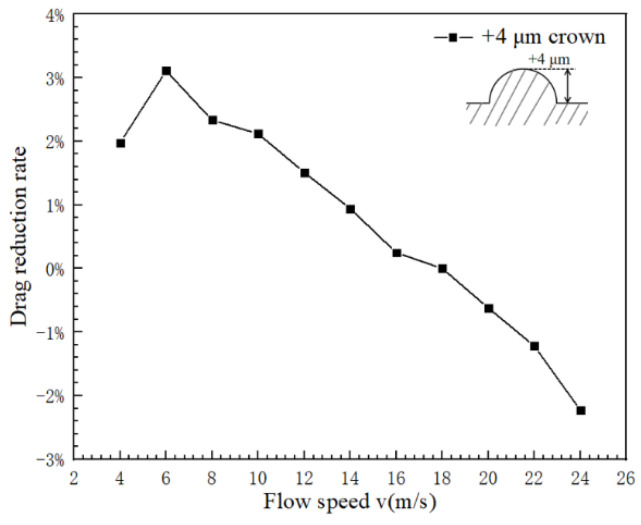
Relationship between surface drag reduction rate and velocity of hemispherical positive coronary process.

**Figure 5 micromachines-12-00059-f005:**
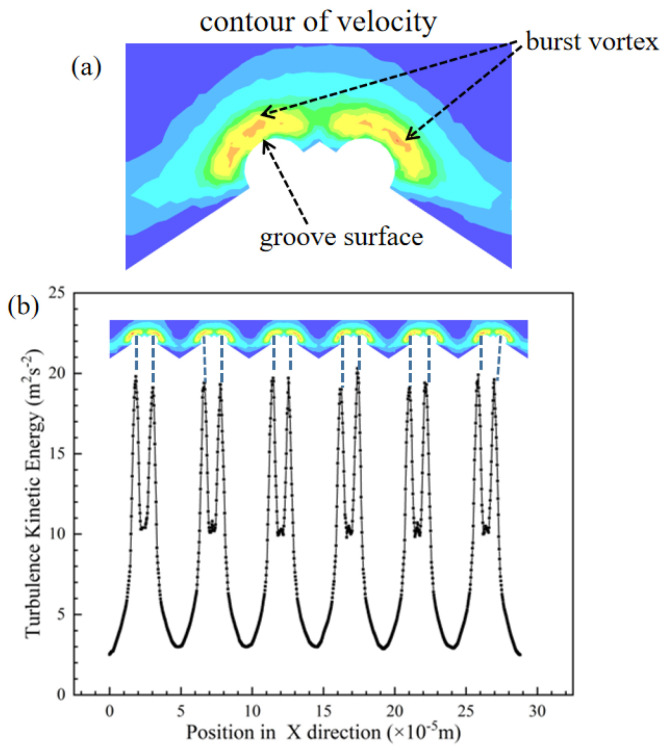
(**a**) Contour of velocity near the groove surface. (**b**) Curve of turbulent kinetic energy on the surface.

**Figure 6 micromachines-12-00059-f006:**
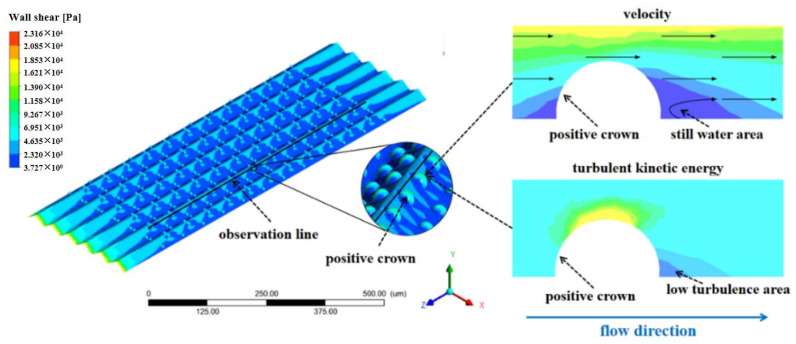
The flow through the positive crowns can form a still water area. In this area the intensity of turbulent kinetic energy is low.

**Figure 7 micromachines-12-00059-f007:**
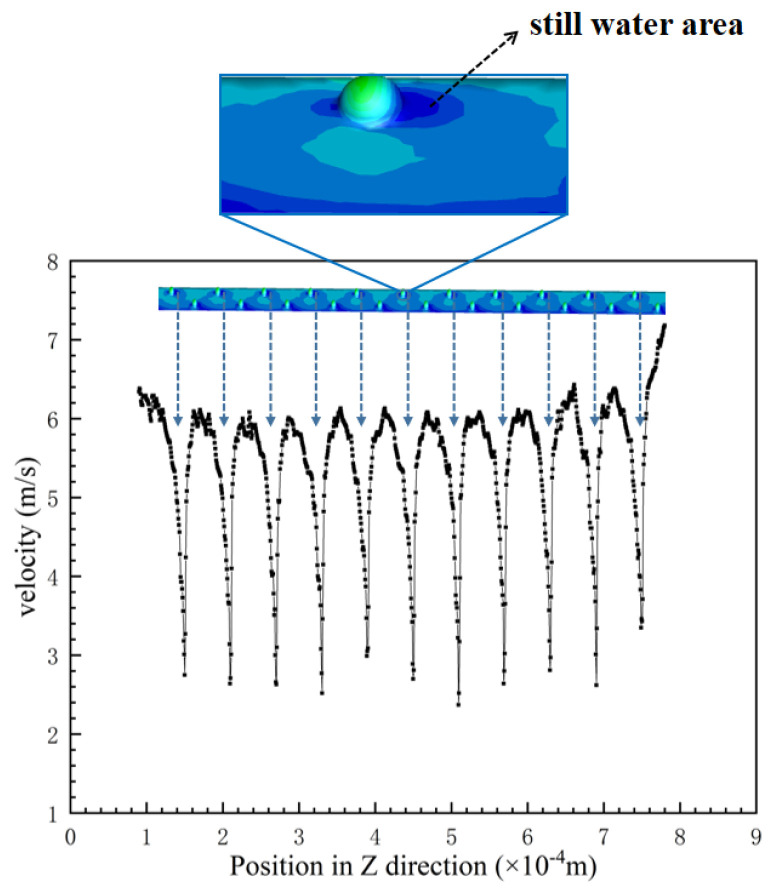
The velocity on the groove surface, and partially enlarges the contour of one crown.

**Figure 8 micromachines-12-00059-f008:**
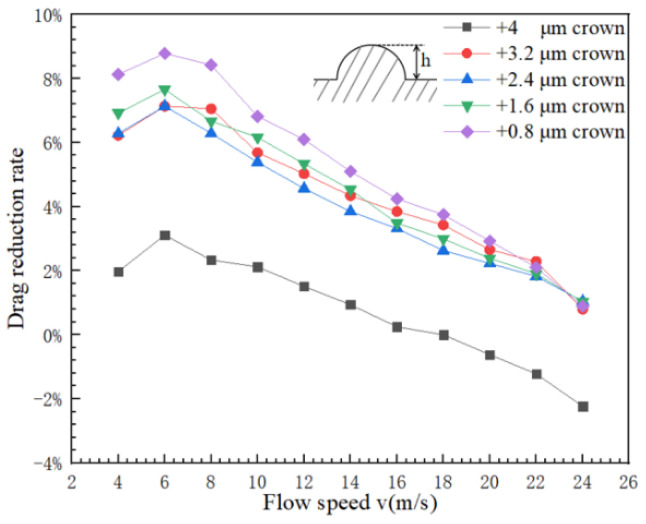
Relationship between the water flow velocity and the drag reduction rate of the groove surface of the positive crown with different heights.

**Figure 9 micromachines-12-00059-f009:**
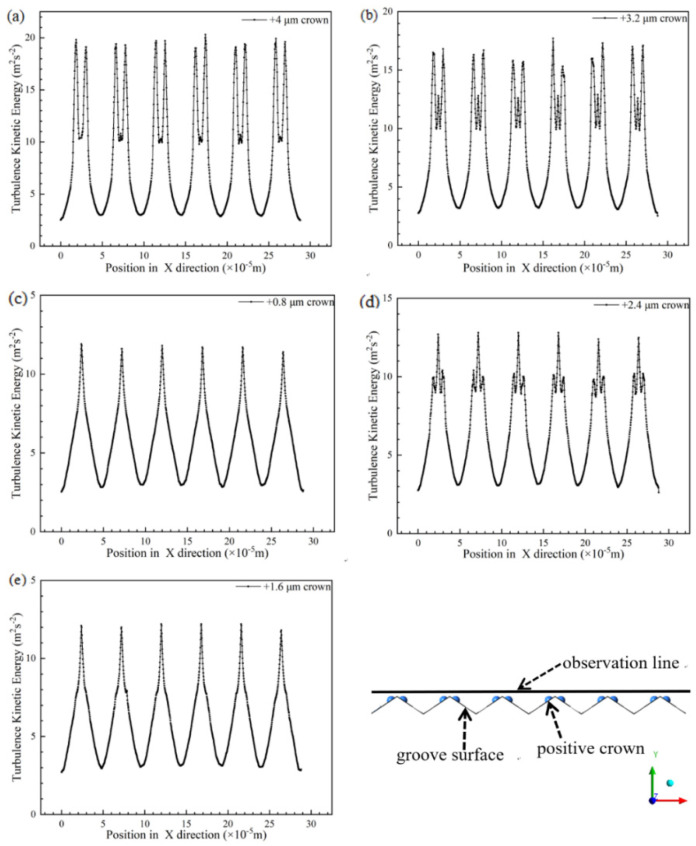
The turbulent kinetic energy which perpendicular to flow direction. From (**a**–**e**), the height of the positive spherical crown changes from +4 μm to +0.8 μm.

**Figure 10 micromachines-12-00059-f010:**
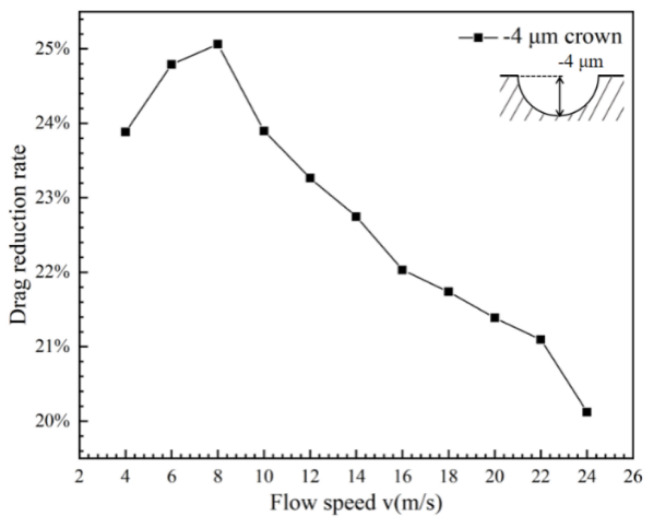
Relationship between surface drag reduction rate and flow velocity of hemispherical negative coronal protrusions.

**Figure 11 micromachines-12-00059-f011:**
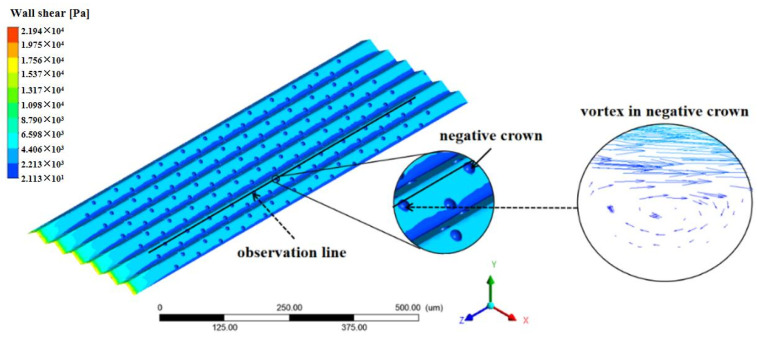
The wall sheer stress of negative crown microstructure at 12 m/s, and the vortex in negative crown.

**Figure 12 micromachines-12-00059-f012:**
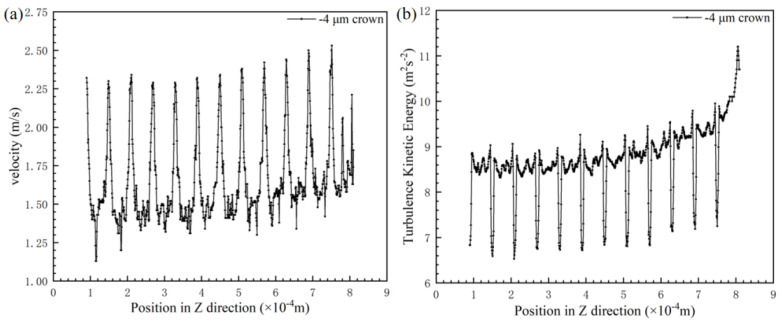
(**a**) Flow velocity curve of the surface of the negative spherical cap structure along with flow velocity, and (**b**) a variation of the turbulent flow energy intensity of negative spherical crown surface.

**Figure 13 micromachines-12-00059-f013:**
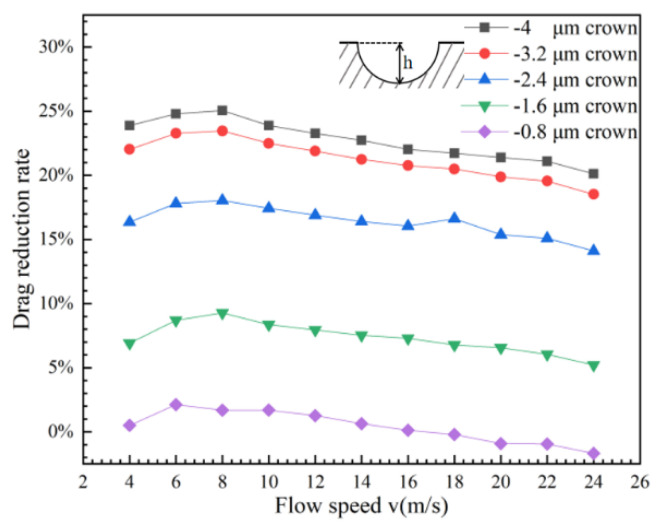
The relationship between the water flow rate and the drag reduction rate of the groove surface of the negative crown with different depths.

**Figure 14 micromachines-12-00059-f014:**
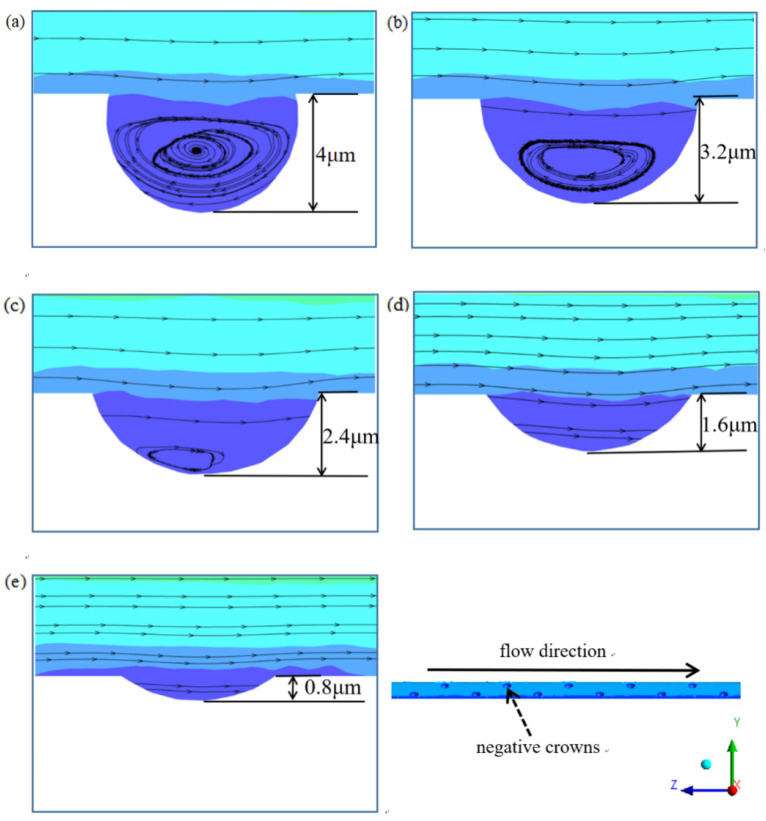
From (**a**–**e**) the longitudinal height of the negative crown is arranged in order from deep to shallow. It can be seen that as the longitudinal height decreases, the size of vortexes in the crown gradually decreases. Surface drag reduction.

**Figure 15 micromachines-12-00059-f015:**
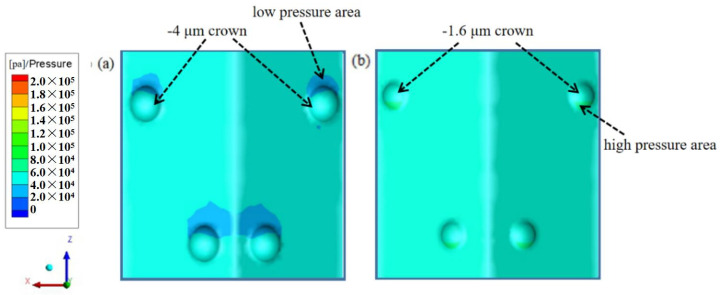
The contour of pressure (**a**) at the depth of −4 μm (**b**) at the depth of −1.6 μm.

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
