# Peer review of "Investigation of the Turbulent Drag Reduction Mechanism of a Kind of Microstructure on Riblet Surface"

_micromachines, 2021, doi:10.3390/mi12010059_

Round 1

Reviewer 1 Report

This is a nice work where a fluid-dynamic model was applied to study the turbulent drag reduction mechanism of microstructures on surfaces. Such a study is timely and would suit the broad readership of chemical engineering.

In particular, the authors demonstrate that in small volumes, the surface drag reduction rate of spherical crown microstructures protrudes in opposing directions. Unfortunately, there are not enough references to the recent contributions within the field of flow chemistry and microreactor engineering, which the authors mention as a possible application field of their study. Therefore, the introduction needs to be modified to cite such works (see, for example, "Green Chem. 2020, 22, 3748-3758", "Ind. Eng. Chem. Res. 2019, 58, 10164", and "ChemCatChem 2018, 10, 3786"). Also, there are some minor English mistakes in the manuscript, which renders the work difficult to read. I would recommend (if possible) rechecking the English with a native speaker.

Author Response

Thanks for your comments. The references recommended by the reviewers have played a very inspiring role in our research on the practical application of microfluidics. After relevant investigation this paper decides to add more references about the application of microfluidic.

Reviewer 2 Report

In this work, the authors theoretically discussed fluid motion on the V-shaped groove surface with spherical crown microstructures by adopting the RNG k-epsilon model. The results showed that the drag reduction rate of negative spherical crown was larger than that of positive spherical crown. They proposed that the vortexes inside the crown can reduce the wall surface resistance. Overall, the manuscript is well written and contains rational explanations. I am pleased to support its publication if the authors are willing to address the following comments:

  1. To gain confidence in the author’s model the numerical approach, I would suggest providing model verification into the main text or supplementary material. Also, it would be better to give an illustration to specify the associated boundary conditions, which are important in the turbulent model, in the computational domain.
  1. In eqs 1-5, some variables and parameters need to be defined when first used.
  1. In Figure 2, the heights of crowns were different, and serially decreased. Did the authors compare the present figuration with the case of crows having an identical height?
  1. The reviewer is curious about the influence of space between the crowns on the friction resistance. Is there an optimal distance to have a better surface drag reduction?

Author Response

1. As for model verification:

It is generally believed that because the valley bottom of the groove can effectively limit the turbulence of the fluid flows and reduce the energy exchange between the fluid and the wall, the V-shaped groove can achieve a good drag reduction effect. However, due to the vortices, which formed from the fluid flows through the tip of the V-shaped groove, interact with the fluid inside the groove. The drag reduction effect of the entire structure will weaken. So, in this work, how to accurately simulate the influence of the vortices at the bottom of the groove on the turbulence is very important.

The RNG k-ε is a model derived from the instantaneous Naiver-stokes equation with the method of re-normalization group [1]. Compared with other turbulence models, the RNG k-ε model has the following advantages:

(1)The RNG k-ε model has a more accurate description of the dissipation rate of turbulent kinetic energy

(2)The RNG k-ε model considers the influence of eddy currents on turbulence and improves the accuracy of vortices flow

(3)The RNG k-ε model provides an analytical formula for the turbulent Prandtl number, which can more accurately simulate the problem of turbulent boundary layer flow

Therefore, the RNG k-ε model is suitable for the relevant scenarios simulated in this article.

As for an illustration to specify the associated boundary condition:

It has been changed in figure 3 in the main text.

2. Thanks for your reminder, the parameters have been added in the main text.

3.Thank for pointing out this problem. This is where the writing of this article is not rigorous. The original Figure 2 in this manuscript wanted to show the difference between ten different kinds of three-dimensional spherical crown models. In fact, the height of the crowns in each three-dimensional spherical crown model is the same. The reworked manuscript has modified the original figure and explained in the caption

4.    The dimensions of the V-shaped groove in this article are referenced from the published paper [2]. In the reference, researchers measured the actual scales with a stylus surface profiler, obtained space size of real shark retlets. Based on these measurements, this paper did a simple avenging processing and designed the spacing of the V-shaped grooves and crowns in this model.

The reviewer's proposal on the influence of space size on drag reduction rate is a very constructive research idea. We will be further studied in this way.

Reference:

[1] G. Mompean. Numerical simulation of a turbulent flow near a right-angled corner using the Speziale non-linear model with RNG K–ε equations. Computers & Fluids 1998, 27(7): 847-859.   https://doi.org/10.1016/S0045-7930(98)00004-8

[2] Zhao DY, Tian QQ, Wang MJ, Huang ZP, Wang T. Experimental study of hydrophobic mechanism of micro-riblets on imitative shark skin. Journal of Dalian University of Technology 2013, 53(04): 503-507.
